# CREB overexpression in dorsal CA1 ameliorates long-term memory deficits in aged rats

Xiao-Wen Yu[1], Daniel M Curlik II[1,2], M Matthew Oh[1], Jerry CP Yin[3], John F Disterhoft[1]*

[1]Department of Physiology, Northwestern University,Feinberg School of Medicine, Chicago, United States; [2]Department of Behavioral Sciences, Psychology Program, York College of Pennsylvania, York, United States; [3]Departments of Genetics and Neurology, University of Wisconsin-Madison, Madison, United States

**Abstract** The molecular mechanisms underlying age-related cognitive deficits are not yet fully elucidated. In aged animals, a decrease in the intrinsic excitability of CA1 pyramidal neurons is believed to contribute to age-related cognitive impairments. Increasing activity of the transcription factor cAMP response element-binding protein (CREB) in young adult rodents facilitates cognition, and increases intrinsic excitability. However, it has yet to be tested if increasing CREB expression also ameliorates age-related behavioral and biophysical deficits. To test this hypothesis, we virally overexpressed CREB in CA1 of dorsal hippocampus. Rats received CREB or control virus, before undergoing water maze training. CREB overexpression in aged animals ameliorated the long-term memory deficits observed in control animals. Concurrently, cells overexpressing CREB in aged animals had reduced post-burst afterhyperpolarizations, indicative of increased intrinsic excitability. These results identify CREB modulation as a potential therapy to treat age-related cognitive decline.

**\*For correspondence:**
jdisterhoft@northwestern.edu

**Competing interests:** The authors declare that no competing interests exist.

## Introduction

Age-related cognitive impairments are observed across multiple species, including laboratory rodents and humans. Forms of learning that require an intact hippocampal formation, such as spatial navigation, are severely impacted in aged humans (*Etchamendy et al., 2012*), rats (*Gallagher and Pelleymounter, 1988*), and mice (*Bach et al., 1999*). Although age-related cognitive deficits have been observed across numerous tasks and species, not all aged subjects display these impairments (*Gallagher and Pelleymounter, 1988*; *Knuttinen et al., 2001a*, *2001b*). Therefore, the aging population can be split into aged individuals who are cognitively-impaired, and others who are cognitively-unimpaired. These cognitively-unimpaired 'super agers' are capable of learning and remembering at young-like levels (*Curlik et al., 2014*; *Gallagher and Pelleymounter, 1988*; *Knuttinen et al., 2001a*; *Rogalski et al., 2013*).

Identifying the molecular mechanisms that differentiate successful from unsuccessful cognitive-agers is highly desirable, as knowledge of the underlying mechanisms will greatly facilitate treatment of these impairments. One likely mechanism contributing to age-related cognitive deficits is a decrease in the intrinsic excitability of CA1 pyramidal neurons. Numerous studies have revealed that CA1 pyramidal neurons from aged animals have reduced intrinsic excitability when compared to those from young animals. Specifically, pyramidal neurons from area CA1 of the dorsal hippocampus of aged animals exhibit a larger post-burst afterhyperpolarization (AHP) than those from young animals (*Disterhoft and Oh, 2007*, *2006*; *Gant et al., 2006*; *Landfield and Pitler, 1984*; *Oh et al.,*

2013). The magnitude of this age-related decrease in neuronal excitability correlates with age-related cognitive deficits (*Tombaugh et al., 2005*). Aged impaired (AI) animals have larger AHPs than both young animals, and aged unimpaired (AU) animals. Interestingly, the AHP amplitude from AU animals is no different than that of young animals (*Matthews et al., 2009*; *Moyer et al., 2000*; *Tombaugh et al., 2005*). Moreover, pharmacological compounds that reduce the amplitude of the AHP in vitro, ameliorate age-related cognitive impairments in vivo (*Kronforst-Collins et al., 1997*; *Moyer et al., 1992*; *Oh et al., 1999*). Based on these findings, we are searching for molecular pathway(s) that modulate both cognition and intrinsic cellular excitability.

One such pathway is activated by the transcription factor, cAMP response element-binding protein (CREB; (*Alberini, 2009*). Numerous studies have manipulated CREB activation in young animals to demonstrate its essential role in memory formation (*Bernabeu et al., 1997*; *Bourtchuladze et al., 1994*; *Dash et al., 1990*; *Deisseroth et al., 1998*; *Kaang et al., 1993*). Memories for spatial and cued information were impaired in transgenic mice expressing a dominant negative form of CREB (*Pittenger et al., 2002*). Likewise, mutations which prevent CREB from being activated by inhibiting its phosphorylation also impaired memory (*Kida et al., 2002*). Conversely, increases in CREB activity via transgenic or viral means facilitated memory. For instance, expressing a partially-active form of CREB (VP16-CREB) in the amygdala resulted in stronger memories for contextual and cued fear conditioning (*Viosca et al., 2009b*). Likewise, infusion of HSV-CREB into hippocampus or amygdala, results in CREB overexpression and facilitation of memory for the Morris water maze (*Sekeres et al., 2010*), water cross maze (*Brightwell et al., 2007*), or fear conditioning (*Josselyn et al., 2001*) in young animals.

Notably, manipulations that increase CREB activity in young animals have also been found to increase intrinsic excitability of their neurons. For example, expression of VP16-CREB resulted in increased neuronal excitability of numerous brain regions, including CA1 of hippocampus, the locus coeruleus, the nucleus accumbens, and the amygdala (*Dong et al., 2006*; *Han et al., 2006*; *Lopez de Armentia et al., 2007*; *Viosca et al., 2009b*). Moreover, overexpression of wild-type CREB has also been shown to be sufficient to enhance excitability (*Yiu et al., 2011*; *Zhou et al., 2009*). Taken together, these previous studies in young adult animals indicate that increasing CREB activity facilitates memory and increases intrinsic neuronal excitability.

Contrary to young adults, very little is known about age-related changes to hippocampal CREB levels and/or activity. Furthermore, the few published studies which have examined CREB levels in aging appear to present conflicting results. One study reported that levels of total CREB protein were unchanged with aging, however, levels of CREB phosphorylated at S133 (pCREB) decreased with age (*Foster et al., 2001*). Another study reported a decrease in total CREB with aging, which was specifically observed in aged animals that had impairments in spatial memory (*Brightwell et al., 2004*). A third study found that total CREB was unchanged, while pCREB increased with age (*Monti et al., 2005*). While the exact nature of age-related CREB levels is unclear, these findings do suggest that CREB levels in the hippocampal formation change with age. Additionally, upstream activators of CREB, such as cAMP, have been found to be decreased with age (*Bach et al., 1999*). Activation of protein kinase A, the next step in this activation cascade, is required for the learning-induced reduction in post-burst AHPs (*Oh et al., 2009*; *Zhang et al., 2013*), and reduced in aged rats (*Karege et al., 2001*). Calcineurin, a phosphatase that dephosphorylates CREB, has also been found to be increased with age (*Foster et al., 2001*). All of these factors could result in a decrease in CREB activation, which led us to hypothesize that increasing CREB levels would ameliorate both the cognitive and biophysical deficits observed in normal aging subjects. We tested this hypothesis by using an adeno-associated viral vector to overexpress wild-type rat CREB in CA1 of dorsal hippocampus. Spatial learning and memory were then assessed using two different Morris water maze protocols; a less challenging version, and a more difficult version. Our results revealed that CREB overexpression in dorsal CA1 ameliorated age-related long-term biophysical and behavioral deficits.

## Results

### Infusion of AAV-CREB caused CREB overexpression in infected CA1 neurons from young and aged rats

While several studies have utilized viral vectors to achieve overexpression of CREB in brains of young animals (*Josselyn et al., 2001*; *Sekeres et al., 2010*; *Zhou et al., 2009*), few studies have done the same in aged animals. Thus, we characterized the infection efficiency and the extent of CREB overexpression of our new adeno-associated viral vector. AAV-CREB virus with a GFP reporter (*Figure 1—figure supplement 1*) was bilaterally injected into CA1 region of 5 young (3–4 months old, mo), and four aged (29–30 mo) male, Fischer 344XBrown Norway F1 hybrid rats. Two weeks after the viral injections, animals were transcardially perfused, and their brains post-fixed to make coronal sections (40 μm) for immunofluorescent staining. Three sections from each animal were stained for NeuN and GFP, to determine the percentage of neurons that were GFP-positive (GFP+): i.e., cells that had been infected by AAV-CREB virus. By visualizing the GFP signal, we observed widespread infection through the medial-lateral axis of CA1 (*Figure 1a*), with approximately 2 mm of spread in the anterior-posterior axis. Over 95% of cells in this infected area were positive for GFP, and there was no difference in the percentage of GFP+ cells between young (95.24 ± 0.71%) and aged (96.30 ± 0.83%) animals ($t_7 = 1.26$, n.s.; *Figure 1b*). To confirm that viral infection led to overexpression of CREB, another set of sections from the beginning of the infected area, with a lower infection rate to encompass sufficient number of GFP- cells, was stained for CREB and GFP (*Figure 1c*). The average CREB intensity of GFP- neurons was set at 100%, and the CREB immunocytochemical signal from GFP+ neurons was normalized to that value. One sample t-tests against a hypothetical average of 100% revealed that GFP+ cells expressed more CREB than GFP- cells in both young (151.4 ± 5.16%, $t_4 = 9.97$, p=0.0006) and aged (141.0 ± 3.44%, $t_3 = 11.91$, p=0.0013) animals, and no differences were observed in CREB expression of GFP+ cells between young and aged animals ($t_7 = 1.59$, n.s.; *Figure 1d*). Together, these data indicate that our AAV-CREB virus infects the same percentage of neurons and achieves the same amount of CREB overexpression in both young and aged animals.

### AAV-CREB ameliorated age-related spatial memory deficits; less challenging water maze

To determine whether viral overexpression of CREB in dorsal CA1 facilitated learning and memory, we injected dorsal CA1 of young and aged rats with AAV-CREB (7 young and 15 aged) or a control AAV-GFP virus (7 young and 17 aged) which lacked the sequence for CREB (*Figure 1—figure supplement 1*). We used approximately double the number of aged animals to account for their increased variability, due to the presence of both AI and AU animals. Two weeks after AAV infusion, the rats were trained on a less challenging protocol for Morris water maze. Animals first received training with the visible platform procedure to ensure that they did not have sensorimotor deficits that would prevent them from performing the task. On the visible platform day, animals were trained with eight trials, and animals were able to successfully acquire the task. A repeated measures ANOVA, with age and virus as the main factors, and trial as the repeated factor, revealed significant effects of trial ($F_{7, 42} = 15.77$, p<0.0001) and of age ($F_{1, 42} = 7.67$, p=0.0084), but not of virus ($F_{1, 42} = 0.001$, n.s.). All animals successfully reached the visible platform on the last trial, and therefore no animals were excluded from further training (*Figure 2a*, inset).

Three days after visible platform training, animals were trained with the hidden platform task. All animals were trained with 5 trials of the hidden platform task per day, over four consecutive days (*Figure 2a*). Each animal's average cumulative proximity to the hidden platform during each training session was used to assess performance. We found that young animals acquired the task more quickly than aged animals, but there were no differences amongst viral groups for either age group. A repeated measures ANOVA, with age and virus as the between groups measures, and session as the repeated measure, revealed significant main effects of session ($F_{3, 42} = 63.3$, p<0.0001) and of age ($F_{1, 42} = 14.72$, p=0.0004), but not of virus ($F_{1, 42} = 0.14$, n.s.). We observed that on the last session of hidden platform training, aged GFP animals tended have higher cumulative proximities than the other groups. We examined these differences on the last session with a two-way ANOVA, with age and virus as the between groups factors. This revealed a significant main effect of age ($F_{1, 42} =$

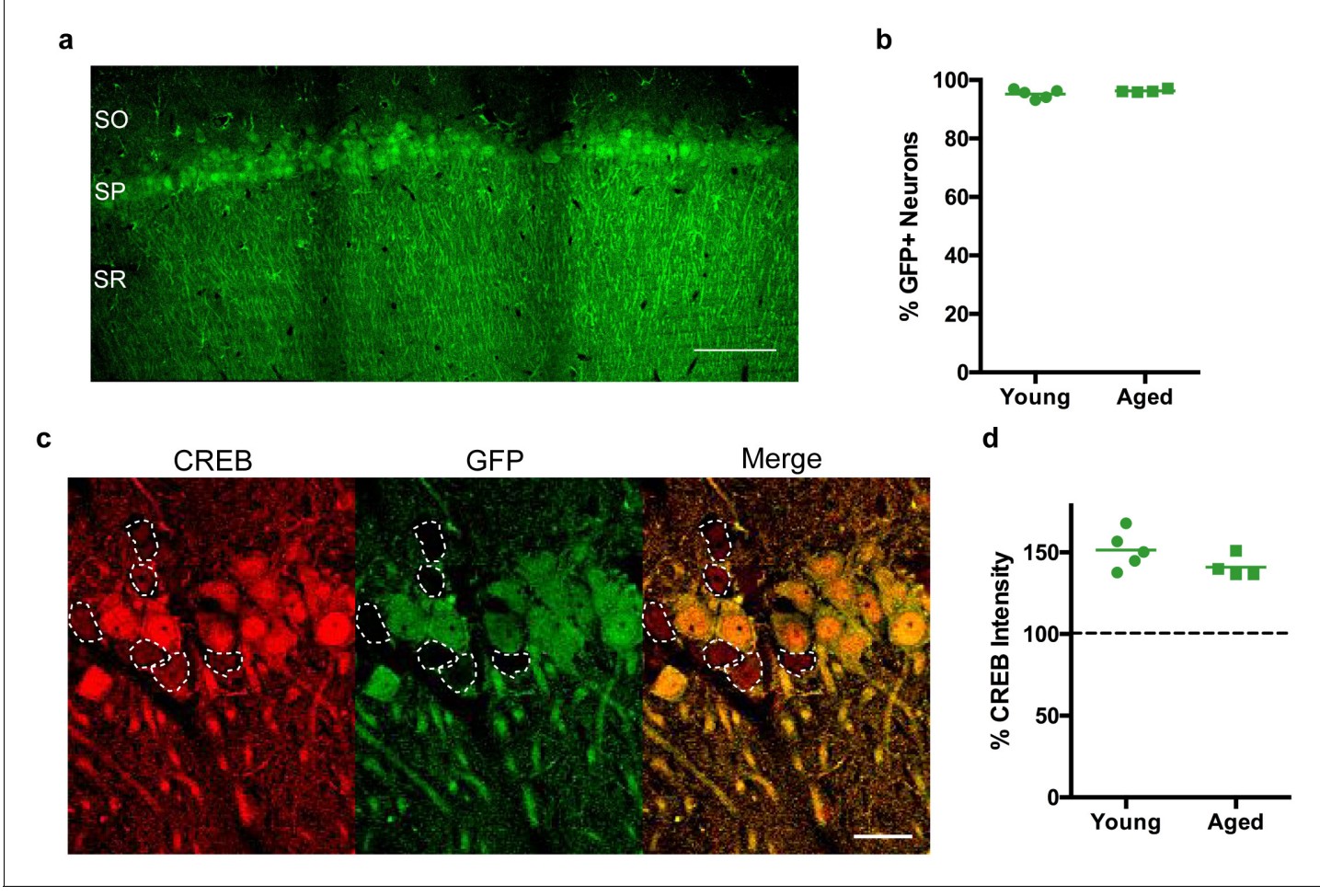

**Figure 1.** CREB expression levels are increased in CA1 neurons of both young and aged rats following stereotaxic injection of AAV-CREB vector into dorsal CA1 region. (a) The AAV-CREB vector contained a GFP reporter. Example of a stitched 40x confocal image of GFP staining in CA1 area. SO – stratum oriens; SP – stratum pyramidale; SR – stratum radiatum. Scale bar = 100 μm. (b) The percentage of infected CA1 neurons in young and aged animals (n = 5, 4) was quantified by dividing the number of GFP positive cells by the total number of NeuN positive cells for each animal. (c) Example of CREB immunofluorescence in infected (green) and uninfected (not green, outlined) cells. Note the higher intensity in GFP positive cells. Scale bar = 20 μm. (d) AAV-CREB infected cells (i.e., GFP positive cells) had higher CREB expression than uninfected cells. Relative CREB expression level of infected cells was quantified as a percentage of CREB intensity of uninfected cells (normalized to be 100%, dashed line) in young and aged animals (n = 5, 4).

The following figure supplement is available for figure 1:

**Figure supplement 1.** Schematic for viral constructs.

9.70, p=0.003), but not of virus ($F_{1,42}$ = 1.16, n.s.). Post-hoc comparisons (Tukey's) of cumulative proximity on the last day of training confirmed that aged GFP animals had greater cumulative proximities than both young GFP animals (p=0.031), and young CREB animals (p=0.023).

To assess memory for the platform location, all animals underwent probe trials 1 hr after the end of each daily hidden platform session, as well as 1 day and 4 days after the last session of training (*Figure 2b*). Percent time in the target quadrant during the first 20 s of each probe trial was used as our measure of spatial memory. A repeated measures ANOVA, with age and virus as main factors, and probe trial as the repeated factor, revealed significant main effects of trial ($F_{5,42}$ = 31.66, p<0.0001) and age ($F_{1,42}$ = 23.63, p<0.0001), as well as a trial by age interaction ($F_{5,42}$ = 2.61, p=0.026). Together, these results indicate that aged animals did not perform as well as young animals on probe trials. Since young animals treated with GFP and CREB performed comparably across all sessions of visible platform training ($F_{1,12}$ = 0.50, n.s.), hidden platform training ($F_{1,12}$ = 1.17, n.

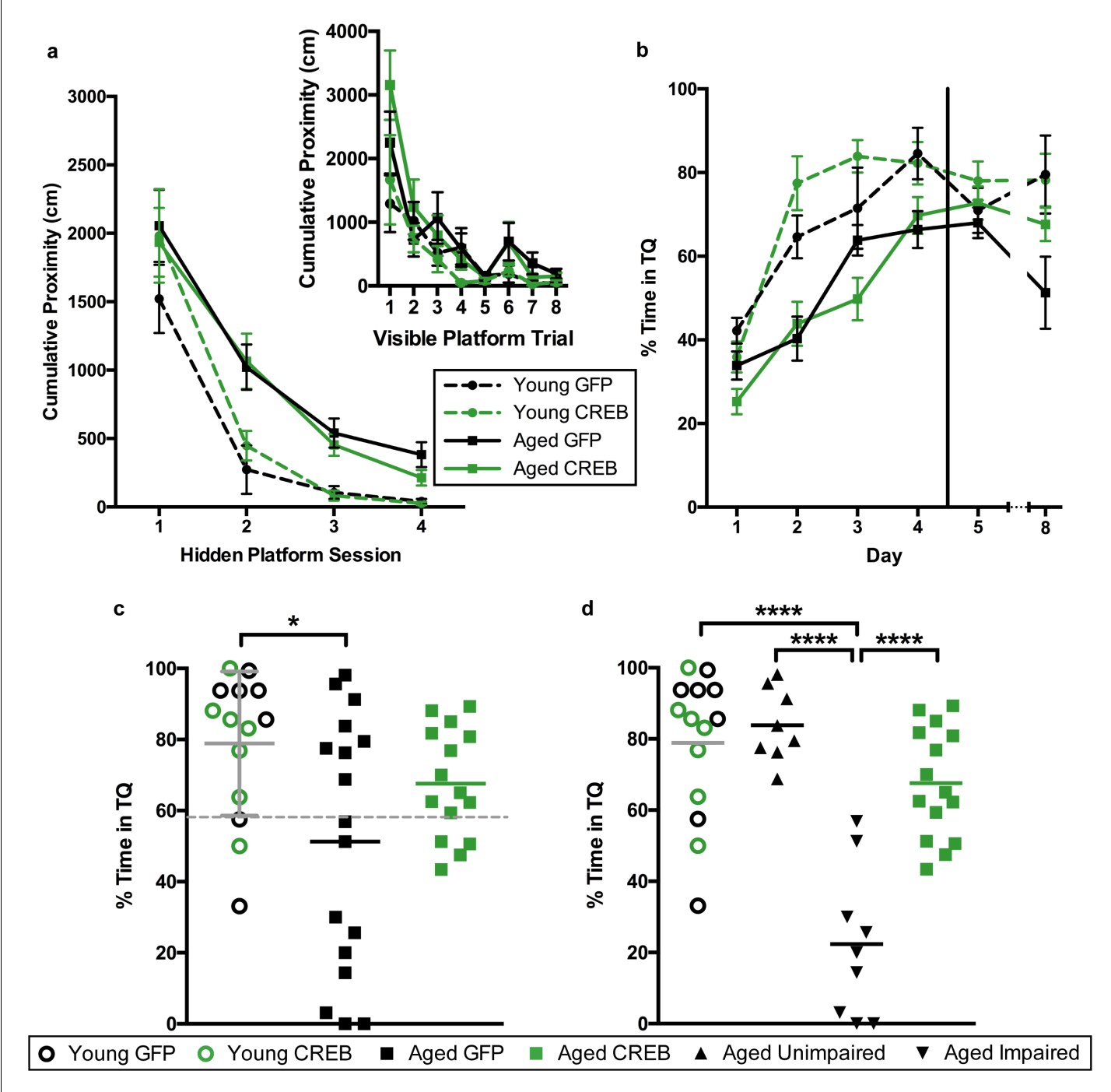

**Figure 2.** AAV-CREB ameliorates spatial memory deficits in aged animals. (a) Performance on visible platform over eight trials (inset), and hidden platform over 4 sessions of training (five trials each session; one session per day) for young animals and aged animals that received control AAV-GFP, or AAV-CREB virus (n = 7, 7, 17, 15). All animals were capable of learning the visible and hidden platform task. While young animals performed better than aged on hidden platform training, no virus-based differences were observed. (b) Performance of young and aged animals that received AAV-GFP or AAV-CREB on probe trials run 1 hr after each session of hidden platform training, as well as 1 day and 4 days after the end of training (n = 7, 7, 17, 15). A difference was found between young and aged animals, but no effect of virus was observed. (c) Aged rats given control AAV-GFP vector were significantly impaired on memory for the platform, as measured by a probe test conducted 4 days after the end of hidden platform training. Dashed line indicates the average performance of young rats minus 1 SD. (d) Aged rats classified as aged impaired (AI: n = 9) performed significantly worse on the probe test conducted 4 days after the end of the hidden platform training as compared to young adult (n = 14), aged unimpaired (AU: n = 8) and aged rats given AAV-CREB (n = 15). No significant differences were observed between young adult, AU and aged CREB rats. Results in a and b represent mean ± SEM, horizontal bars in c and d represent mean, c represents mean ± SD or mean only.

s.), and probe tests ($F_{1, 12} = 0.39$, n.s.) we collapsed the data from these animals into one 'young' group for further analyses.

As upregulation of CREB has been shown to enhance long-term memory, we more closely examined performance during the last probe test, conducted four days after the last session of hidden platform training. A one-way ANOVA comparing performance between young, aged GFP, and aged CREB animals was significant ($F_{2, 43} = 4.45$, p=0.018). Tukey's multiple comparisons test revealed that young animals spent significantly more time in the target quadrant compared to aged GFP (p=0.01), but not compared to aged CREB (n.s., *Figure 2c*). When a D'Agostino and Pearson omnibus normality test was run, we found young and aged CREB data to have a Gaussian distribution, while aged GFP did not (young n.s, aged CREB n.s., aged GFP p=0.045). To determine if we had two groups within the aged GFP group, we separated the animals into two groups; AU and AI, using [young average - one standard deviation] as the cut off (grey dotted line, *Figure 2c*). The two new aged GFP groups were each found to have Gaussian distributions (AU n.s., AI n.s.). Based upon these new behavioral groupings, we reanalyzed the day four probe data, and found a significant difference between the groups (one-way ANOVA: $F_{3, 42} = 23.78$, p<0.0001, *Figure 2d*). Tukey's multiple comparisons test revealed that the AI group was significantly different from each of the other groups: young (p<0.0001), AU (p<0.0001), and aged CREB (p<0.0001), while young, AU, and aged CREB animals were not different from each other. These results indicate that in aged animals, AAV-CREB injection ameliorated the long-term memory deficits seen in AAV-GFP animals, to the point where their performance was no different to that of young animals.

## Behavioral effects were due to CREB overexpression in CA1 only

Two to three weeks after the end of behavioral testing, animals were euthanized to collect tissue for RNA, protein, and to gather biophysical data. To verify that viral overexpression of CREB was limited to the CA1 sub-region, the three major hippocampal sub-regions were separated to examine their CREB mRNA levels. A two-way ANOVA revealed a main effect of virus on CREB mRNA levels in CA1 ($F_{1, 42} = 13.20$, p=0.0008, *Figure 3a*). Bonferroni's multiple comparisons test indicated that AAV-CREB animals had significantly more CREB mRNA than those that had received control AAV-GFP virus in both young (p=0.018), and aged animals (p=0.037). We examined RNA from dentate gyrus and CA3 in a subset of animals. In dentate gyrus, a two-way ANOVA revealed no effect of virus ($F_{1, 16} = 0.065$, n.s.), but did find an effect of age ($F_{1, 16} = 5.87$, p=0.028, *Figure 3b*), indicating aged animals had less CREB mRNA in dentate gyrus, regardless of which virus they received. In CA3, no effect of age ($F_{1, 16} = 0.73$, n.s.) or virus ($F_{1, 16} = 0.36$, n.s.) was found for CREB mRNA levels (*Figure 3c*). These results indicate that any behavioral effects seen in these animals could be attributed to overexpression of CREB specifically in the CA1 sub-region of the hippocampus.

As hypothesized, injection of AAV-CREB rescued behavioral deficits in aged, but not young adult rats. However, it was not known if differences in basal level expression of CREB might underlie the behavioral differences. Therefore, CREB mRNA and protein levels were compared between the AU and AI animals (*Figure 3—figure supplement 1*). Surprisingly there was no significant group difference in CREB mRNA levels between the AU and AI animals (p>0.05). However, there was a significant positive correlation between CREB protein levels and performance on the last day of training (probe 4: *Figure 3—figure supplement 2*; r = 0.78, p=0.0002). These data suggest that aged animals with higher basal levels of CREB protein may be better at learning a hippocampus dependent task than those with lower basal levels of CREB protein.

## AAV-CREB infection reduced AHP amplitudes in aged neurons

In addition to the increase in CREB mRNA levels, we hypothesized that CREB overexpression would result in increased intrinsic excitability in the CA1 pyramidal neurons infected by AAV-CREB (*Yiu et al., 2011*; *Zhou et al., 2009*). Whole-cell current-clamp recordings were made from CA1 pyramidal neurons held near −69 mV. Specifically, the postburst AHP was measured following a train of 15 suprathreshold current injections. We compared three different groups of cells from young and aged animals; AAV-CREB infected cells (CREB+), their neighboring uninfected cells (CREB−), and cells from AAV-GFP injected animals. Two-way ANOVAs with main factors of age and cell type revealed similar changes in both the peak and slow AHPs: significant effect of cell type ($F_{2, 63} > 4$, p's < 0.05), but not of age ($F_{1, 63} < 1$, n.s.) (*Figure 3—figure supplement 3*, *Supplementary file*

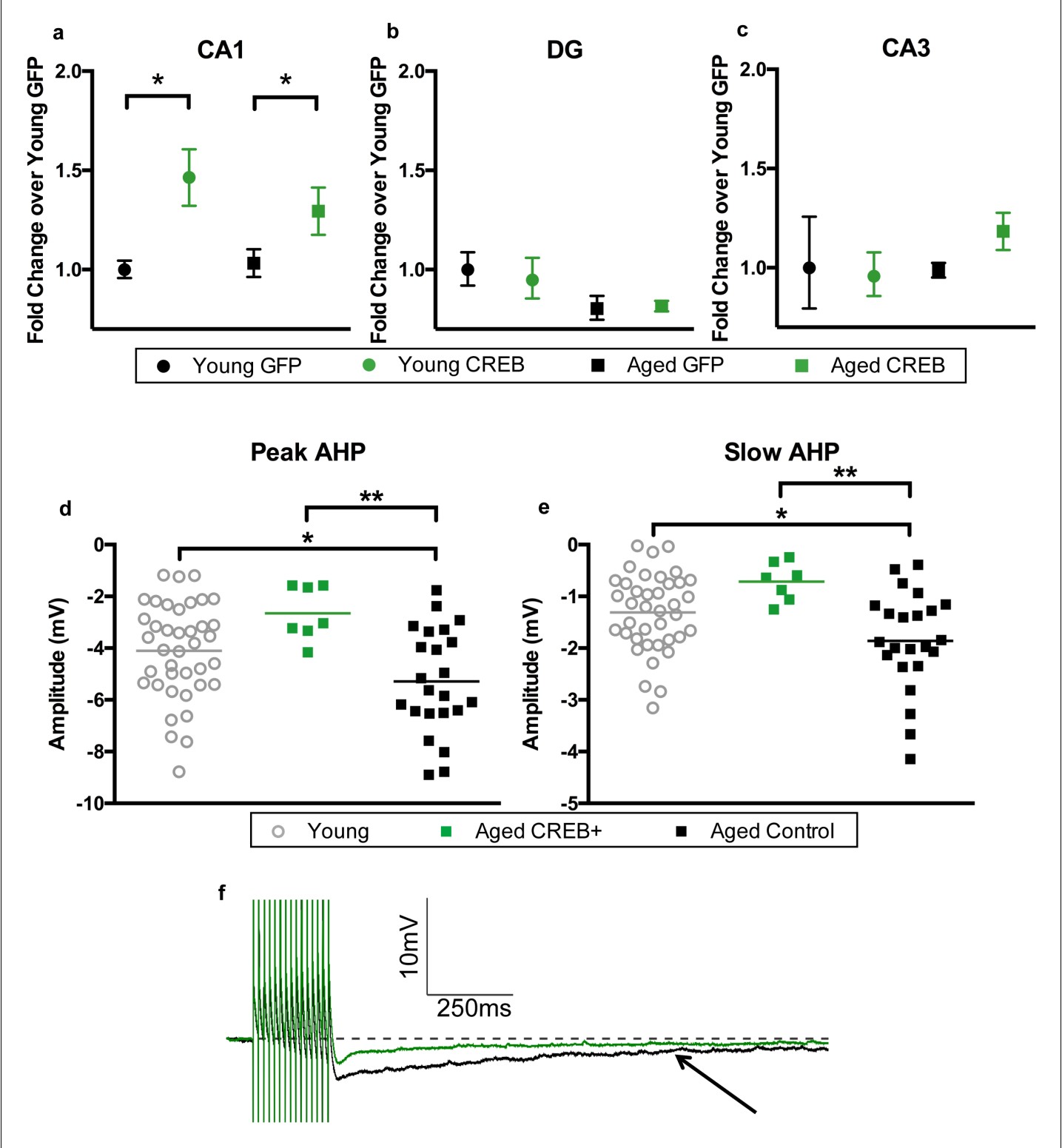

**Figure 3.** AAV-CREB results in higher CREB mRNA levels in CA1, and reduces AHP in infected aged animals. CREB mRNA levels relative to young GFP group in a) CA1, (b) DG, and c) CA3. In both young and aged animals, AAV-CREB injected animals had more CREB mRNA in CA1 (n = 7, 7, 17, 15). This viral difference was not observed in DG (n = 4, 6, 4, 6) or CA3 (n = 4, 6, 4, 6). (d) Peak postburst AHP is significantly reduced in CA1 pyramidal neurons (n = 7) from aged AAV-CREB animals as compared to control cells (n = 23) from aged animals. No significant differences were observed in peak postburst AHP between CA1 neurons from young adult (n = 39) and aged AAV-CREB animals. (e) Similarly, the slow postburst AHP from CA1 neurons of young adult and aged AAV-CREB animals were significantly reduced as compared to control cells from aged animals. (f) Example postburst AHP

*Figure 3 continued on next page*

*Figure 3 continued*

traces from aged CREB+ (green) and aged control (black) CA1 pyramidal neurons. Arrow indicates 1 s time point where slow postburst AHP was measured. Results in a, b, c represent mean ± SEM.

The following figure supplements are available for figure 3:

**Figure supplement 1.** Aged unimpaired and impaired animals had the same amount of CREB mRNA.

**Figure supplement 2.** CREB protein levels are positively correlated with probe trial performance in aged GFP animals.

**Figure supplement 3.** Viral infection state had no effect on postburst AHP size in young CA1 pyramidal neurons, but did affect aged neurons.

*1a,b*). Tukey's multiple comparisons test revealed that aged CREB+ cells had significantly smaller peak and slow AHPs than both aged CREB- and aged GFP cells. Importantly, no differences were found between any of the young cell types (*Figure 3—figure supplement 3a and b*), or between aged CREB- and GFP (*Figure 3—figure supplement 3c and d*). Therefore, postburst AHP data for all young cells were combined into one group, and the postburst AHP data for aged CREB- and GFP cells were combined into one aged control group. Analyses revealed significant differences of the postburst AHP in the three groups of cells: one-way ANOVA for peak AHP, $F_{2, 66} = 6.22$, p=0.003; for slow AHP $F_{2, 66} = 6.54$, p=0.003 (*Figure 3d,e*). Furthermore, Tukey's multiple comparisons test revealed that the postburst AHP in aged controls (peak, $-5.29 \pm 0.42$ mV: slow, $-1.86 \pm 0.20$ mV) were significantly larger than those from young (peak, $-4.10 \pm 0.30$ mV: slow $-1.31 \pm 0.12$ mV) and aged CREB+ (peak, $-2.65 \pm 0.39$ mV: slow, $-0.71 \pm 0.14$ mV) cells. Importantly, young and aged CREB+ were not statistically different from each other. Passive membrane properties (such as input resistance and resting membrane potential) were not significantly different across groups of neurons (*Supplementary file 2a,b*). These data indicate that in aged animals, cells infected with AAV-CREB were found to be significantly more excitable than control cells, to the extent where they were no different from young cells.

## AAV-CREB ameliorated age-related spatial memory deficits; more difficult water maze

Unlike previous studies (*Brightwell et al., 2007*; *Josselyn et al., 2001*; *Sekeres et al., 2010*), we did not observe any difference in water maze performance between our young AAV-GFP and AAV-CREB injected animals. In past studies, animals were trained on difficult versions of the tasks, where control animals (i.e., those given control viral injections) displayed little long-term memory, but memory in animals overexpressing CREB was strongly facilitated. In the experiments described in *Figure 2*, we trained the rats over four sessions (one session/day) with five trials per session. This training regimen may have been 'less challenging' and produced a ceiling effect in our young animals, where no further improvements could be observed with increased CREB levels. Therefore, a new group of rats were trained using a more difficult training protocol, to determine whether AAV-CREB could facilitate learning and/or memory in young animals.

Young and aged rats were infused with AAV-CREB (8 young and eight aged) or a control AAV-GFP virus (7 young and 15 aged) in the dorsal CA1 region. Two weeks after AAV infusion, animals were trained with the visible platform procedure, which was performed exactly as we described earlier. Repeated measures ANOVA, with age and virus as the main factors and trial as the repeated factor, revealed significant effects of trial ($F_{7, 34} = 10.61$, p<0001) and of age ($F_{1, 34} = 4.06$, p=0.052), but not of virus ($F_{1, 34} = 0.40$, n.s.), with no significant interactions between any of the factors. All animals successfully reached the visible platform on the last trial, and therefore no animals were excluded from further training (*Figure 4a*, inset).

Three days after visible platform training, animals were trained with the hidden platform task. Rather than train animals with five trials per each daily session, as we did previously, we trained each animal with three trials per day, over four consecutive days (*Figure 4a*). The inter-trial interval (ITI) was also changed. Previously the ITI was $19 \pm 1$ min, and we reduced that to 1 min (*Josselyn et al., 2001*). As expected, young animals acquired the task more quickly than aged animals, however there were no differences amongst viral groups for either age group. A repeated measures ANOVA,

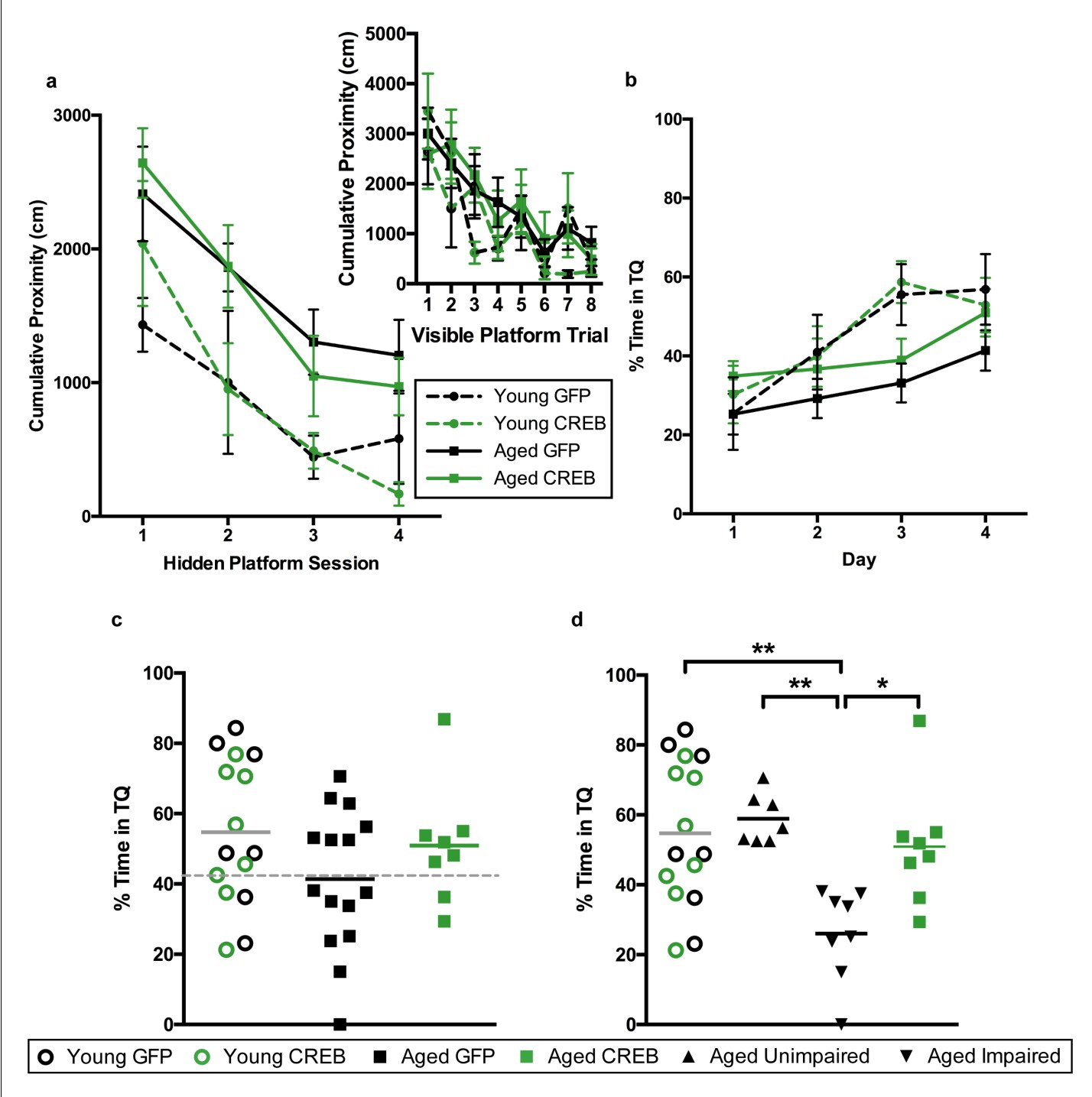

**Figure 4.** AAV-CREB ameliorates age-related spatial memory deficits on a more difficult water maze task. (**a**) Performance on visible platform over eight trials (inset), and hidden platform over 4 sessions of training (three trials each session; one session per day) for young animals and aged animals that received control AAV-GFP, or AAV-CREB virus (n = 7, 8, 15, 8). All animals were capable of learning the visible and hidden platform task. While young animals performed better than aged on hidden platform training, no virus-based differences were observed. (**b**) Performance of young and aged animals that received AAV-GFP or AAV-CREB on probe trials run 1 hr after each session of hidden platform training (n = 7, 8, 15, 8). A difference was found between young and aged animals, but no effect of virus was observed. (**c**) Aged rats given control AAV-GFP vector trended towards worse performance on a probe test conducted 1 hr after the last hidden platform training session. Dashed line indicates the average performance of young rats that successfully learned minus 1 SD. (**d**) Aged rats classified as aged impaired (AI: n = 8) performed significantly worse on the probe test conducted on day 4 of hidden platform training as compared to young adult (n = 15), aged unimpaired (AU: n = 7) and aged rats given AAV-CREB

*Figure 4 continued on next page*

*Figure 4 continued*

(n = 8). No significant differences were observed between young adult, AU and aged CREB rats. Results in a and b represent mean ± SEM, horizontal bars in c, d represent mean.

with age and virus as the between groups measures, and session as the repeated measure, revealed significant main effects of session ($F_{3, 34}$ = 26.24, p<0.0001) and of age ($F_{1, 34}$ = 11.85, p=0.0015), but not of virus ($F_{1, 34}$ = 0.001, n.s.), with no significant interactions between any of the factors.

To assess memory for the platform location, all animals underwent probe trials 1 hr after the end of each daily hidden platform session (*Figure 4b*). A repeated measures ANOVA, with age and virus as main factors, and probe trial as the repeated factor, revealed significant main effects of trial ($F_{3, 34}$ = 13.05, p<0.0001), but not of age ($F_{1, 34}$ = 3.31, n.s.) or virus ($F_{1, 34}$ = 0.85, n.s.). However, a trial by age interaction was observed ($F_{3, 34}$ = 3.15, p=0.028). Together, these results indicate that aged animals did not perform as well as young animals across the probe trials. Since young animals treated with GFP and CREB performed almost identically across all sessions of visible platform training ($F_{1, 7}$ = 2.039, n.s.), hidden platform training ($F_{1, 13}$ = 0.03, n.s.), and probe trials ($F_{1, 13}$ = 0.01, n.s.) we once again collapsed the data from these animals into one 'young' group for further comparisons against aged animals.

As with the first set of animals, we split the aged GFP group into AU and AI. The cut off (grey dotted line, *Figure 4c*) was determined using young animals' probe trial performance on the last day [young average – one standard deviation]. Two young animals (one GFP and one CREB) did not successfully learn the task, i.e. performed below chance on this probe trial, and were therefore omitted from this calculation. Upon reanalysis of the probe four data, we found a significant difference between the groups (one-way ANOVA: $F_{3, 34}$ = 6.36, p<0.002, *Figure 4d*). Tukey's multiple comparisons test revealed that the AI group was significantly different from each of the other groups [young (p=0.0024), AU (p=0.0034), and aged CREB (p=0.028)], while young, AU, and aged CREB animals were not different from each other. Despite being trained on a more challenging protocol for water maze the young CREB group still performed comparably to the young GFP group. However, in aged animals, AAV-CREB injection ameliorated long-term memory deficits seen in the AAV-GFP group, to the point where their performance was no different to that of young animals.

To confirm this protocol was more challenging, we compared the behavioral performance by groups on the two different versions. Given that we saw a significant effect of age, but not of virus for both tasks, the young were collapsed into one group, and the aged were collapsed into a second group for each version. A two-way ANOVA (training protocol as the between groups measure; hidden platform training session as the repeated measure) revealed significant main effects of session ($F_{3, 81}$ = 36.89, p<0.0001) and of training protocol ($F_{1, 27}$ = 4.36, p=0.047). This indicates that as expected, young animals found it more difficult to acquire the task when trained on the more difficult training protocol. The same effects were found for performance on probe trials; a two-way ANOVA revealed a significant main effect of probe trial ($F_{3, 81}$ = 44.00, p<0.0001), and of training protocol ($F_{1, 27}$ = 18.37, p=0.0002). Aged animals also performed worse on hidden platform training and probe trials. A two-way ANOVA revealed a significant main effect of hidden platform training session ($F_{3, 159}$ = 55.28, p<0.0001), and of training protocol ($F_{1, 53}$ = 24.40, p<0.0001). A two-way ANOVA for probes revealed a significant main effect of probe trial ($F_{3, 159}$ = 26.93, p<0.0001), and of training protocol ($F_{1, 53}$ = 21.35, p<0.0001). These results reveal that both young and aged animals exhibited worse performance on both the learning and memory components of the more difficult water maze task, indicating that the lack of AAV-CREB facilitation of young animals' performance was not due to a ceiling effect.

One hour after the final probe trial, the animals were euthanized to collect tissue for RNA. To verify that CREB overexpression occurred, the CA1 sub-region was examined for CREB mRNA levels. A two-way ANOVA revealed a main effect of virus on CREB mRNA levels in CA1 ($F_{1, 34}$ = 84.24, p<0.0001). Bonferroni's multiple comparisons test indicated that AAV-CREB animals had significantly more CREB mRNA than those that had received control AAV-GFP virus in both young (AAV-CREB 2.34 ± 0.27 fold change over AAV-GFP 1.05 ± 0.09; p<0.0001), and aged animals (AAV-CREB 2.23 ± 0.18 fold change over AAV-GFP 1 ± 0.07; p<0.0001). These data indicate that as shown

previously, the behavioral effects seen in this second group of aged animals could also be attributed to the overexpression of CREB.

## Discussion

This is the first study to directly increase CREB levels in aged animals, and as a result, rescue their age-related cognitive deficits. Administration of AAV-CREB resulted in widespread infection of dorsal CA1 in young and aged rats, which specifically increased CREB mRNA levels in dorsal CA1. Importantly, this increase in CREB was sufficient to ameliorate age-related long-term spatial memory deficits. Furthermore, as predicted, aged pyramidal neurons infected by AAV-CREB had reduced post-burst AHPs, indicating that infected cells were more excitable. These data reaffirm that aging-related processes increase the AHP in CA1 neurons, and leads to decreased excitability that contributes to memory impairment. Increasing CREB levels is sufficient to reverse these processes and normalize memory function. Our data is clear, and is consistent with some previously reported work. However, it is at odds with other reports, and we suspect that a number of different experimental variables contribute to this inconsistency in the literature.

Aged animals, as a group, displayed robust spatial memory deficits. However, not every aged animal was impaired. Nearly all aged CREB animals, and a subset of aged GFP rats were able to learn and remember both the less challenging, and more difficult task at young-like levels. These results are consistent with those of earlier studies, indicating that approximately half of aged rats are impaired on this task (*Disterhoft and Oh, 2006*; *Gallagher et al., 1993*). With that said, our aged control animals could be separated into those with impaired long-term spatial memory, and those without impaired long-term spatial memory, whereas aged animals receiving AAV-CREB were much more homogenous, consistently performing at, or near, young-like levels. We found water maze performance by our young and, presumably AU animals were not improved with AAV-CREB, which is supported by results of a prior study which revealed that viral administration of CREB-Y134F improved fear memory in young 'poor learners', with no improvement in young 'good-learners' (*Cowansage et al., 2013*). These findings are also supported by a study by *Brightwell et al. (2004)*, which found that AI animals trained on water maze had lower levels of CREB protein than young animals and AU animals (2004). Together, these previous findings and our present results suggest that lower levels of CREB protein in AI animals contributes to their cognitive deficits, and increasing CREB in these animals can rescue their behavioral deficits. This rescue by CREB overexpression was most likely mediated by increased levels of pCREB. Despite reports of decreased CREB activation with aging, via increases in calcineurin (*Foster et al., 2001*) and decreases in PKA (*Karege et al., 2001*), we hypothesize that increasing the absolute levels of their substrate, CREB, would lead to an increase in the levels of pCREB and CREB activity. However, the detailed mechanistic relationship between the levels of CREB, pCREB, and CREB activity is unlikely to be straight forward (*Briand et al., 2015*).

A second parameter that might contribute to experimental variability is the nature of the CREB protein that is overexpressed. We found that CREB overexpression increased the intrinsic excitability of aged CA1 neurons. The AHPs from AAV-CREB infected neurons were reduced when compared to neighboring uninfected cells, or GFP-injected controls, as previously reported (*Yiu et al., 2011*; *Zhou et al., 2009*). These results mirror those from studies using pharmacological methods where treatment that reduced the AHP also facilitated cognitive performance in aged animals (*Deyo et al., 1989*; *Kronforst-Collins et al., 1997*; *Moyer et al., 1992*; *Oh et al., 1999*; *Weible et al., 2004*). Interestingly, we found that CREB overexpression had no observable effect on the size of AHPs in young neurons. Previous studies using partially-active CREB-Y134F or VP16-CREB have demonstrated their ability to facilitate long-term potentiation and reduce the AHP in cells from young animals (*Lopez de Armentia et al., 2007*; *Suzuki et al., 2011*; *Yu et al., 2016*). The increase in downstream transcription is likely to be much greater when a partially activated form of CREB, as opposed to wild-type, is used. This may in part explain why we did not see any effect of virus on AHP amplitude in cells from young animals.

A third variable is the excitability state of the animal at the time of training. Studies using CREB overexpression in young animals have produced mixed results. *Zhou et al. (2009)* used HSV-CREB to overexpress CREB in amygdala of mice, which resulted in a smaller AHP amplitude when measured 300 ms after the final action potential, as compared to neighboring control cells. Peak AHP

was not found to be different (2009). Another study transfected primary mice hippocampal neurons to overexpress CREB and found that transfected cells had increased excitability (*Yiu et al., 2014*). Lastly, when HSV-CREB was injected into locus coeruleus of rats, excitability was unchanged, as compared to rats injected with control virus (*Han et al., 2006*). These varying results may be due to differences in slice vs. cultured neurons, mouse vs. rat tissue, or even due to differing brain regions, but we suspect that the 'excitability state' of the animals is a major uncontrolled determinant. In our hands, CREB overexpression was not sufficient to cause any differences in peak or slow AHP amplitude in CA1 neurons from young animals. Although we saw similar levels of CREB overexpression in both age groups, only the aged animals exhibited a change at the biophysical level. It is possible that while the young animals are already at an 'optimal state', and have sufficient excitability or plasticity for learning and memory, the aged animals are in a 'sub-optimal state', and therefore have room for improvement. This effect has been demonstrated before, where systemic application of L-type calcium channel blocker nimodipine (*Deyo et al., 1989*) or cholinesterase inhibitor galantamine (*Weible et al., 2004*) was beneficial for aged animals, but had no effect on young. Similarly, exposure to environmental enrichment, which can indirectly lead to increased CREB activation (*Hu et al., 2013*), reduced the AHP in neurons from aged animals, but not young (*Kumar and Foster, 2007*). For potential therapeutic purposes, this may be a desirable trait, so as to only affect those who have room for improvement.

Given our biophysical results, it is not surprising that AAV-CREB (vs. AAV-GFP) injections had no effect on any aspect of young animals' water maze performance. We have demonstrated that viral overexpression of CREB was achieved in these young animals, both at the mRNA and protein levels. Overexpression of wild-type rat CREB, as opposed to a mutated form of CREB, may partly explain the lack of behavioral effect. Expression of VP16-CREB in young animals has led to transient facilitation of eye-blink conditioning (*Gruart et al., 2012*), as well as facilitation of memory for contextual and cued fear (*Viosca et al., 2009a*). Additionally, expression of a CREB-Y134F in young animals has given enhanced memory for both fear conditioning and water maze (*Restivo et al., 2009*; *Suzuki et al., 2011*). These studies suggest that the increase in CREB expression we observed with our AAV-CREB was not sufficient (when compared to that achieved by mutated forms of CREB) to facilitate the behavioral performance of young animals. However, some studies have shown facilitation of fear conditioning as a result of HSV overexpression of wild-type CREB in the amygdala (*Josselyn et al., 2001*; *Sekeres et al., 2010*). In these studies, animals underwent weak training, where control animals failed to form a memory, but animals overexpressing CREB were able to do so. In the current study, we initially used a less challenging version of the water maze task, and delivered 4 sessions of training with five trials per session. We suspected this 'strong' training may have produced a ceiling effect in our young animals, where increasing CREB activity could not lead to further improvements. Thus, we injected a new group of animals with AAV-GFP and AAV-CREB, and trained them on a more difficult version of the task, to determine if young CREB animals' performance would then be facilitated. The difficult version did prove more challenging, with both young and aged animals taking longer to learn the task, and spending less time in the target quadrant across probe trials as compared to the less challenging version. Interestingly, the pattern of behavioral results was very similar to that observed during the less challenging task: CREB mediated rescue in aged rats, but had no effect in young rats. This suggests that the difference between the viral overexpression of HSV and AAV may be at play. HSV is known to give strong, but transient gene expression, while AAV tends to provide more subtle, but stable expression over long periods of time (*Neve et al., 2005*). These data indicate that strong CREB overexpression, such as in the case of HSV, or a mutant form of CREB is required to achieve behavioral facilitation in young adult animals, given that they are already at an 'optimal learning state'.

Notably, CREB mRNA levels were not related to behavioral performance. Young GFP and CREB groups whose behavior did not differ showed differing CREB mRNA levels; while AU and AI groups, who by definition differ in behavior, had the same levels of CREB mRNA. However, levels of CREB protein and pCREB have repeatedly been shown to be related to behavior (*Brightwell et al., 2004*; *Cowansage et al., 2013*; *Josselyn et al., 2001*; *Pittenger et al., 2002*; *Viosca et al., 2009a*). Importantly, we provide evidence that basal level of CREB protein is correlated to successful learning in aged rats (*Figure 3—figure supplement 2*). Together, these findings suggest that CREB mRNA and protein levels are not directly related to each other. Moreover, while CREB mRNA levels served to confirm viral overexpression, only CREB protein levels were predictive of behavioral performance.

Interestingly, CREB mRNA levels did not differ between young and aged animals in CA1, but it was reduced in DG of aged animals. While this is an important observation, given the lack of relationship between CA1 CREB mRNA levels and behavior, our focus continues to be on the CA1 sub-region. We chose to target this sub-region of the hippocampus as it has exhibited biophysical deficits with age (*Bach et al., 1999*; *Disterhoft and Oh, 2007*; *Tombaugh et al., 2005*). Additionally, CA1 has been targeted to successfully facilitate memory in young animals with CREB overexpression (*Sekeres et al., 2010*). In our present study, targeting the CA1 sub-region has successfully enhanced intrinsic excitability of aged pyramidal neurons, with a corresponding amelioration of age-related memory deficits.

The postburst AHP is influenced by numerous manipulations. One of these is successful learning of a behavioral task, where animals that learn exhibit smaller AHPs than those who do not (*Oh and Disterhoft, 2015*; *Moyer et al., 1996*). However, the learning-related reduction in postburst AHP is transient, and AHP amplitudes return to baseline within two weeks after the end of behavioral training (*Moyer et al., 1996*; *Thompson et al., 1996*; *Tombaugh et al., 2005*). All biophysical recordings in this study were made between two to three weeks after the end of behavioral testing in order to measure the baseline AHP, and to avoid any learning-induced changes in the AHP. However, this still raises the question of how CREB overexpression resulted in the reduction of the AHP. CREB may modulate the AHP by indirectly influencing the levels or activity of AHP channels via one or more of CREB's downstream transcriptional targets. Alternatively, it is possible that CREB could act on AHP channels via a previously unknown non-transcriptional mechanism. Elucidating this mechanism would be of great interest. Despite not fully understanding the mechanisms involved, the consequences of CREB's enhancement of neuronal excitability have been examined. Neurons that have increased CREB activity become more excitable (*Lopez de Armentia et al., 2007*), and are then preferentially recruited to form new memories (*Han et al., 2007*; *Yiu et al., 2014*; *Zhou et al., 2009*).

While previous studies have manipulated factors upstream of CREB, such as cAMP levels (*Bach et al., 1999*), this is the first study to directly test CREB's role in age-related cognitive deficits. Previous studies have shown increases in calcineurin with aging (*Foster et al., 2001*), decreases in PKA, (*Foster et al., 2001*; *Karege et al., 2001*), and AI animals have been shown to have reduced CREB protein (*Brightwell et al., 2004*). All of these could result in decreases in CREB function, but only one study has manipulated CREB levels with aging. *Mouravlev et al. (2006)* did a longitudinal study revealing that overexpressing wild-type CREB in young rats prevents age-related cognitive deficits from arising later in life. However, it was still unclear whether CREB would be able to reverse the age-related deficits after they arose. In our present study, CREB overexpression alone was sufficient to rescue the long-term memory deficits in aged animals. We hypothesize that CREB's effects on behavioral performance may have been via the normalization of the previously reported deficits in CREB activity/function. This hypothesis is supported by our finding that CREB protein levels strongly correlate with performance on probe 4, where aged animals with the lowest levels of CREB protein performed the worst. This suggests that aged animals with low CREB protein levels are more likely to be AI. On the other hand, aged animals with high CREB protein levels, either naturally or due to AAV-CREB infusions, are more likely to perform at young-like levels. Alternatively, it is also possible that CREB overexpression may have compensated for age-related deficits via other mechanisms. Possibilities include CREB's ability to modulate synaptic plasticity (*Benito and Barco, 2010*), and to induce preferential recruitment of neurons to form a new memory via increasing intrinsic excitability (*Han et al., 2007*; *Yiu et al., 2014*; *Zhou et al., 2009*).

In addition to these findings contributing to a better understanding of the aging process, other studies have shown that CREB and its activation are also negatively impacted in Alzheimer's disease. Amyloid –beta treatment reduces CREB levels and can cause it to be retained in the cytoplasm (*Arvanitis et al., 2007*; *Pugazhenthi et al., 2011*), while high BACE-1 levels reduce CREB activation (*Chen et al., 2012*), and CREB-mediated gene expression is aberrant in brains of Alzheimer's patients (*Satoh et al., 2009*). A recent study has also shown that Tau accumulation reduces CREB activity by increasing calcineurin levels (*Yin et al., 2016*). Together, these results suggest that impaired, or sub-optimal CREB function in aging may make aged individuals more vulnerable to Alzheimer's disease. Furthermore, this indicates that CREB may not only be targeted for amelioration of cognitive deficits seen in normal aging, it could also be an important pharmacological target in the prevention or treatment of Alzheimer's disease.

# Materials and methods

## Subjects

Young adult (3–4 mo) and aged (29–30 mo) male F1 hybrid Fischer 344XBrown Norway (F344XBN) rats were used for this study. All animals were obtained from the National Institute on Aging colony at Charles River Laboratories (Raleigh, NC). All rats were housed in a temperature-controlled facility with a 14 hr light/10 hr dark cycle and allowed free access to food and water. They were group housed, and allowed to acclimate in the Northwestern University vivarium for a minimum of one week prior to any experimentation. All procedures were performed in strict accordance with the recommendations in the Guide for the Care and Use of Laboratory Animals of the National Institutes of Health. All animals were handled according to protocols approved by Northwestern IACUC (Protocol number: IS00002081, Animals Welfare Assurance: A3283-01) following NIH guidelines. All surgery was performed under isofluorane anesthesia, and every effort was made to minimize suffering.

## Viral vectors

An adeno-associated viral vector (AAV) was used to overexpress rat CREB (AAV-CREB). The construct encoded the sequence for endogenous rat CREB, downstream of a chicken $\beta$-actin promoter with a CMV enhancer. This was followed by a GFP sequence downstream from an internal ribosomal entry site. A second construct (AAV-GFP), lacking the CREB sequence served as a control (vectors were a generous gift from Dr. Corinna Burger). Both were cloned into shuttle vectors and packaged into AAV serotype nine by Virovek (Hayward, CA). The final viral titres were $10^{13}$ viral particles/mL. Rats were randomly assigned to receive AAV-CREB or control AAV-GFP.

## Stereotaxic injections

All surgeries were performed using sterile procedures. Animals were anesthestized with isofluorane gas, first in an induction chamber, then while secured in a stereotaxic frame (David Kopf Instruments, Tujunga, CA), with lambda and bregma equal in the vertical plane. A thin layer of ophthalmic ointment was applied to keep the eyes moist. Subjects were administered Buprenex sub-cutaneously (0.03 mg/kg) to minimize discomfort. The scalp was incised and retracted, and bilateral holes were drilled through the skull above the hippocampi at the injection sites in dorsal CA1. The injection sites were 5.0 mm posterior and ±3.0 mm lateral from bregma, and 2.1 mm ventral from the dura. Glass micropipettes with calibrated internal diameters were pulled and cut to have 30–45 µm openings. Glass micropipettes were fitted to a blunt-tipped 10 µL Hamilton syringe, mounted to the stereotaxic frame. For each injection, 1 µL of vector was infused into the hippocampus at a rate of 0.1 µL/min using a Stoelting QSI microinjection pump. The glass micropipette was left in place for an additional 5 min before being gradually removed from the brain. After both injections were complete the scalp was sutured, and animals were given Rimadyl sub-cutaneously (5 mg/kg) to alleviate post-surgical discomfort. Rats were allowed to recover under a heat lamp before returning to their home cage, where they remained for two weeks, before the start of any further experiments.

## Perfusions and coronal sectioning

Rats were deeply anesthetized, then transcardially perfused through the left ventricle, first with 0.1 M phosphate-buffer saline (PBS, pH 7.4), then with 4% paraformaldehyde. The brains were extracted, and post-fixed in 4% paraformaldehyde overnight, then stored in PBS until sectioning. After removing the cerebellum and the anterior portion of the brain (to reveal the fornix), 40 µm sections were made through hippocampus on a Leica VT100s Vibratome. Tissue was stored in PBS 4°C.

## Immunohistochemistry

Brain sections were stained using a free-floating immunofluorescence procedure. PBS-X (0.01 M phosphate-buffered saline containing 0.5% Triton-X-100) was used to make up all other solutions, and all incubations were performed at room temperature, unless otherwise stated. Tissue was permeabilized using PBS-X for 10 min then blocked with 5% normal goat serum (Jackson Immunoresearch, West Grove, PA) for 4 hr. Sections were incubated overnight at 4°C with primary antibodies. The next day, sections were washed in PBS-X then incubated with secondary antibody for 2 hr. After further washes, tissue was mounted using ProLong gold antifade (Life Technologies, Carlsbad, CA).

Stereology utilized brain sections 600 µm apart, all within the infected area. Primary antibodies were chicken anti-GFP (1:2000, Abcam, Cambridge, MA, #13970), and mouse anti-NeuN (1:300, Millipore, Temecula, CA, #MAB377). Secondary antibodies were anti-chicken IgG conjugated to AlexaFluor633, and anti-mouse IgG conjugated to AlexaFluor594, both purchased from Life Technologies, and used at 1:500 dilution.

CREB intensity stains utilized brain sections 200 µm apart, on the edge of the infected area (to ensure sufficient number of GFP- cells were included in the analysis). Primary antibodies were chicken anti-GFP (1:2000, Abcam, #13970), and rabbit anti-CREB (1:800, Cell Signaling Technologies, Boston, MA, #9197). Secondary antibodies were anti-chicken IgG conjugated to AlexaFluor633, and anti-rabbit IgG conjugated to AlexaFluor594, both purchased from Life Technologies, and used at 1:500 dilution.

Stained tissue sections were imaged on a confocal microscope (Nikon A1R) in Northwestern University's Centre for Advanced Microscopy. Using a 40x objective, 3 z-stacks from non-overlapping fields of view were collected from CA1 of dorsal hippocampus. Laser and PMT settings were optimized for imaging each slide, but remained constant for all images taken from that slide.

Staining for NeuN and GFP was done to calculate the percent of infected neurons, in the infected area of dorsal CA1 from AAV-CREB-injected animals. For each animal (5 young and four aged), three sections from the center of the infected area (600 µm apart) were analyzed. From these sections, three non-overlapping z-stacks were collected from CA1 of each hippocampus, resulting in a total of a total of 18 z-stacks for each animal. The sum of the collapsed z-stack was used for subsequent analyses. The threshold of NeuN-positive signal was first set to encompass the signal from cells, but not debris or processes. The signal was then subjected to the Watershed function, followed by Analyze Particles to count how many NeuN-positive cells that z-stack contained. A 'selection' was then created from a thresholded GFP signal. This was then applied to the NeuN signal, to count how many of the NeuN-positive cells were also GFP-positive, which was then expressed as a percentage. All image analysis was carried out using NIH's ImageJ software.

Staining for CREB and GFP was done to determine whether AAV-CREB-infected cells express more CREB protein than uninfected cells from the same animal. Three sections from each animal were analyzed (5 young and four aged animals). Each section was from the edge of the infected area and separated by 200 µm. From these sections three non-overlapping z-stacks were collected from area CA1 of each hemisphere of the hippocampus, resulting in a total of 18 z-stacks for each animal (three sections X 3 z-stacks X two hemispheres). The sum of each collapsed z-stack was used for subsequent analyses. The threshold of CREB signal was first set to encompass the signal from cell bodies, but not debris or processes. After subjecting the signal to the Watershed function, the intensity of each cell was measured along with ROI size. CREB intensity was then normalized to ROI area to ensure the intensity measurement was independent of ROI size (to account for potential differences in measured area). Cells were manually separated into GFP-positive (GFP+), and GFP-negative (GFP-) groups. For each z-stack, the average intensity for all GFP+ cells was calculated, as well as the average intensity for all GFP- cells. To quantify CREB levels in GFP+ cells relative to GFP- cells, the average CREB intensity from GFP+ cells was first normalized to the average CREB intensity from GFP- cells (average GFP+ intensity/ average GFP- intensity. This value was then multiplied by 100 to obtain a percentage. Thus, GFP- cells had an average CREB intensity of 100%, and a higher percentage for the GFP+ cells was indicative of an increase in CREB levels.

## Morris water maze: less challenging version

We used the Morris water maze task to assess learning, short-term memory, and long-term memory in seven young animals injected with AAV-GFP, seven young animals injected with AAV-CREB, 17 aged animals injected with AAV-GFP, and 15 aged animals injected with AAV-CREB. All training and probe trials were conducted in a circular pool (180 cm in diameter). The water (24 ± 1°C) was made opaque by the addition of white non-toxic paint. Ten minutes before each day of training or testing animals were individually housed in new cages. At the end of each daily session of water maze training animals were returned to their home cages. All animals were handled on three separate occasions, for at least 3 min each, in the week prior to the start of visible platform training.

We first trained the rats on a visible platform task to ensure they were capable of locating, navigating to, and climbing onto the escape platform. The rectangular escape platform (20.3 cm x 25.4 cm) was placed in the center of 1 of 4 quadrants of the pool, with the bottom of the platform sitting

at the level of the water. No distinctive cues surrounded the pool, while the platform had several high-contrast visual cues attached to it. At the start of each trial, the animal was placed approximately 7.5 cm away from the wall of the pool, in the center of the quadrant opposite from the platform-containing quadrant. Each animal was given 60 s to navigate to the visible platform. If the animal did not reach the platform after 60 s, the experimenter gently guided the animal to the platform location. The location of the platform changed after every trial, and each animal received eight trials, with an inter-trial interval of $19 \pm 1$ min. This visible platform testing ensured that all animals used in subsequent behavioural experiments had no gross sensorimotor deficits, as they were able to swim, see, and climb onto the visible escape platform. All animals successfully reached the platform during the last trial of visible platform training, and no animals were excluded from further water maze training.

Three days after the conclusion of visible platform training, all animals were trained with a hidden platform task. Distinctive, high-contrast cues were placed on each of the walls surrounding the water maze pool. The circular escape platform (20 cm diameter) was placed in one quadrant, and submerged 2 cm below the water level, so that animals could not see the hidden platform. The location of the platform remained the same throughout hidden platform training. Each animal received 5 trials of hidden platform training per day, over four consecutive days (total 20 trials) with an inter-trial interval of $19 \pm 1$ min. Each trial began by placing the animal in the centre of one of the three quadrants that did not contain the hidden platform. Each animal was allowed 60 s to locate the escape platform. If the animal did not swim to the hidden platform after 60 s, the experimenter gently guided the animal to the platform. Upon reaching the platform, animals remained on the platform for a further 15 s, before being removed by the experimenter. The animals' cumulative proximity to the platform was measured for each training trial, then the average cumulative proximity was calculated for each animal for each of the 4 days of training. We chose to use the cumulative proximity measurement as it has been shown to be sensitive to age-related cognitive impairments (*Gallagher et al., 1993*).

To test for memory for the platform location probe trials were performed 1 hr after each day of hidden platform training, as well as 1 day and 4 days after the last session of hidden platform training. During these probe trials the escape platform was removed from the pool, and each rat was given 60 s to swim in the pool. Memory for the platform location was assessed by quantifying the percent of time each animal spent in the target quadrant of the pool during the first 20 s of the probe test. The target quadrant was the quadrant that contained the hidden platform during training. Similar to the cumulative proximity measure, percent time in target quadrant has been shown to be sensitive to age-related memory impairments (*Foster, 2012*). In order to prevent extinction, a reminder hidden platform trial was conducted $19 \pm 1$ min after each probe trial. WaterMaze software was used for computerized tracking during trials and for subsequent offline analysis (Actimetrics, Wilmette, IL).

## Collection of hippocampal tissue for RT-PCR and biophysical recordings

Two-three weeks after the end of behavioral testing or viral infusion, rats were euthanized to collect hippocampal tissue. Animals were anesthetized with isoflurane and then were rapidly decapitated. Brains were removed and placed into ice-cold artificial cerebrospinal fluid (aCSF, in mM): 124 NaCl, 26 $NaHCO_3$, 2.5 KCl, 1.25 $NaH_2PO_4$, 2 $MgSO_4$, 25 glucose, 2.4 $CaCl_2$, pH 7.4, oxygenated with 95%:5% $O_2$:$CO_2$. A sucrose-aCSF was used for brains from aged rats (in mM): 206 sucrose, 2.5 KCl, 1.25 NaH2PO4, 26 NaHCO3, 0.1 CaCl2, 3 MgSO4, and 25 glucose. Hippocampi were then extracted, and 300 μm transverse sections cut on a Leica VT100s Vibratome. Sections were then used for biophysical measurements, or frozen in RNAlater (Qiagen, Valencia, CA) for RT-PCR experiments, or immediately frozen for Western blot experiments.

## Biophysical recordings

The 300 μm sections from dorsal hippocampus were prepared as described above, then incubated in aCSF for 20 min at 34°C, before being allowed to return to room temperature for at least 40 min. Slices were then transferred to a submersion chamber at 34–35°C. Slices were visualized using an upright Hamamatsu Orca R2 camera. Patch electrodes (5–7 MΩ) were filled with (in mM) 120 KMeSO4, 10 KCl, 10 Hepes, 4 Mg2ATP, 0.4 NaGTP, 10 Na2 phosphocreatine, 0.04 AlexaFluor 594,

and 0.5% neurobiotin, pH adjusted to 7.4 with KOH. No correction was made for a 10 mV liquid junction potential. All measurements were made 5 min after membrane rupture to allow for adequate solution equilibration. Neurons were held near −69 mV, and recordings were obtained using an Axon Digidata 1550A amplifier, and data analyzed using pClamp 10 software (Molecular Devices, Sunnyvale, CA) and custom routines in Matlab. Matlab routines are available upon request. AHP values were observed using a train of 15 action potentials (50 Hz), evoked by direct somatic current injections. The experimenter was blind to the infection status of the neurons during collection and analysis of biophysical data.

In order to confirm infection status of patched cells sections were fixed overnight in 4% paraformaldehyde and then stored in PBS until re-sectioning. To amplify the GFP and neurobiotin signals, tissue was first re-sectioned. Each slice was embedded in 8% agar then re-sectioned into 4–5 70 μm sections. These were permeabilized in PBS-X, and blocked in 5% normal goat serum (Jackson Immunoresearch) for 4 hr. Sections were incubated with chicken anti-GFP primary (1:2000, Abcam, #13970) overnight at 4°C. The next day, the tissue was washed with PBS-X, then incubated with anti-chicken IgG conjugated to AlexaFluor488 (1:375, Jackson Immunoresearch) and Streptavidin conjugated to AlexaFluor594 (1:1000, Life Technologies). After further washes, tissue was mounted using ProLong gold antifade (Life Technologies). Patched cells were examined for the presence of neurobiotin and GFP signal to determine infection status.

## RT-PCR

Four to five hippocampal slices were placed in RNAlater, and frozen. The major hippocampal sub-regions (CA1, CA3, and dentate gyrus) were manually dissected from these slices (*Núñez-Santana et al., 2014*). Under a Stemi DV4 dissecting microscope (Zeiss), and on ice, a straight cut was first made in front of the blades of the dentate gyrus, to isolate the CA3 sub-region from each slice. The dentate gyrus was then peeled away from the CA1-subiculum, and the subiculum severed from CA1 and discarded. Tissues from the sub-regions were then frozen again before further processing. Tissue from CA1, CA3, or DG was first disrupted manually, then with a QiaShredder (Qiagen) column. RNA was subsequently extracted using RNeasy Plus Mini Kit (Qiagen), according to manufacturer's directions. All samples were eluted in 30 uL RNase-free and DNase-free water. All samples were checked for integrity using Biorad's Bioanalzyer (Hercules, CA) and RNA 6000 Pico chips (Biorad). All samples had RNA integrity numbers (RIN) of 7 or higher and therefore none were excluded from analysis. For each sample, equal amounts of RNA were reverse transcribed into cDNA using SuperScript VILO cDNA Synthesis Kit (Invitrogen). To ensure reagents were contaminant-free concurrent controls containing no VILO enzyme were made and carried through the rest of the real-time polymerase chain reaction (RT-PCR) experiment.

All RT-PCR reactions were carried out in 384-well plates. Each well contained sample cDNA, primer and probe against target genes, and SsoAdvanced Universal Probes Supermix (Biorad). Each plate included controls from the cDNA synthesis step, which lacked the reverse transcription enzyme, and wells in which water replaced the cDNA sample. These negative controls were to confirm that no genomic DNA was present, and that no reagents aside from sample cDNA contained template that could be amplified. All samples were run in triplicate, and reactions carried out on a 7900HT (Applied Biosystems, Foster City, CA).

Primers and probes against GAPDH, CREB1, and GFP were purchased from Integrated DNA Technologies. Utilizing GAPDH as the house-keeping gene, the fold change over young GFP animals were calculated for genes of interest using the $\Delta\Delta$Ct method.

| Gene | Forward primer | Probe | Reverse primer |
|------|----------------|-------|----------------|
| GAPDH | CCAGTAGACTCCACGACATAC | CAGCACCAGCATCACCCCATTTG | AACCCATCACCATCTTCCAG |
| CREB | AGCACTTCCTACACAGCCT | ATTCTCTTGCTGCTTCCCTGTTCTTCA | CACTGCCACTCTGTTCTCTA |
| GFP | GAACCGCATCGAGCTGAA | ATCGACTTCAAGGAGGACGGCAAC | TGCTTGTCGGCCATGATATAG |

## Western blotting

Four to five hippocampal slices were immediately frozen on dry ice. While frozen and on ice, CA1 was manually isolated under a Stemi DV4 dissecting microscope (Zeiss) as described above. CA1 tissue was lysed in RIPA buffer (150 mM NaCl, 25 mM Tris, 1 mM EDTA, 0.1% SDS, 1% Triton X-100, and 0.5% sodium deoxycholate) containing protease and phosphatase inhibitors (Pierce, Rockford, IL). Each sample was manually dissociated with a syringe, and shaken at 4°C for 30 min. All samples were then centrifuged at 14,000 g for 15 min. The supernatants were retained and protein concentration measured by BCA assay (Pierce). Fifteen micrograms of each sample was boiled in 2x Laemmli buffer containing 5% $\beta$-mercaptoethanol (Bio-Rad Hercules, CA). Samples were separated using 4–20% Mini Protean TGX gels (Bio-Rad), and transferred to Immobilin-P PVDF membranes (Millipore, Temecula, CA). Blots were probed for CREB (Millipore Cat #04–767, RRID:AB_1586959, 1:5000) and GAPDH (Thermo Fischer Scientific Cat #MA1-16757, RRID:AB_568547 1:40000, Thermo Scientific, Waltman, CA). Anti-rabbit secondary was first used to react with the anti-CREB antibody. After imaging, HRP signal was quenched using 15% $H_2O_2$. An anti-mouse secondary (Jackson) was then used to react with the anti-GAPDH primary. Immunoreactive bands were visualized using a ChemiDoc XRS+ Molecular Imager System with ImageLab Software (Bio-Rad). Offline quantification of reactive bands also used ImageLab software. All gels were loaded with the same reference sample to allow for comparison across gels. All samples were normalized for loading error using GAPDH signal. Pearson's R values were determined to test for strength of correlations between CREB protein levels and behavioral performance.

## Morris water maze: more difficult version

We determined whether the difficulty of the water maze task affected our results by training a new group of animals on a more difficult version of the task. Similar to the first group, we assessed learning and short-term memory in seven young animals injected with AAV-GFP, eight young animals injected with AAV-CREB, 15 aged animals injected with AAV-GFP, and eight aged animals injected with AAV-CREB. Unless otherwise noted, the training conditions remained the same as the previous protocol.

As before, we first trained the rats on 8 trials of a visible platform task (inter-trial interval of 19 ± 1 min) to ensure they were capable of locating, navigating to, and climbing onto the escape platform. All animals successfully reached the platform during the last trial of visible platform training, and no animals were excluded from further water maze training.

Three days after visible platform training, all animals were trained with a hidden platform task. The location of the platform remained the same throughout hidden platform training. Each animal received 3 trials of hidden platform training per day, over four consecutive days (total 12 trials). The inter-trial interval was 1 min. The animals' cumulative proximity to the platform was measured for each training trial, then the average cumulative proximity was calculated for each animal for each of the 4 days of training. We chose to use the cumulative proximity measurement as it has been shown to be sensitive to age-related cognitive impairments (*Gallagher et al., 1993*).

To test for memory for the platform location probe trials were performed 1 hr after each day of hidden platform training. Memory for the platform location was assessed by quantifying the percent of time each animal spent in the target quadrant of the pool during the first 20 s of the probe test. The target quadrant was the quadrant that contained the hidden platform during training. Similar to the cumulative proximity measure, percent time in target quadrant has been shown to be sensitive to age-related memory impairments (*Foster, 2012*). Unlike in the less challenging version of the task, no reminder hidden platform trial was given after each probe trial. WaterMaze software was used for computerized tracking during trials and for subsequent offline analysis (Actimetrics, Wilmette, IL).

One hour after the last probe trial, animals were euthanized to collect tissue for RT-PCR. Hippocampi were isolated and frozen in RNAlater. The CA1 sub-region was dissected and RNA extracted as described above. RT-PCR reactions for CREB were carried out as described above.

## Statistical analysis

Data are presented at mean ± SEM. N is number of animals, except for biophysical experiments, where n is number of cells. To minimize number of animals used, only n = 4–5 animals were used for

the immunofluorescent staining. For behavior, seven young animals were used for each treatment, while approximately double this number of aged animals were used to accommodate for their increased variance due to presence of both aged impaired and aged unimpaired animals (*Curlik et al., 2014*; *Gallagher and Nicolle, 1993*; *Knuttinen et al., 2001a*).

GraphPad Prism version 6 (GraphPad Software, La Jolla, CA) was used for statistical analysis. One sample t-test, two-tailed student t-tests, one-way, and two-way ANOVAs with Tukey's or Bonferroni's multiple comparisons tests were used as appropriate. Normality was tested using D'Agnostino and Pearson omnibus normality test. Three-way repeated measures ANOVAs were run using Statview version 5.0.1. Significance levels were set at p=0.05. Significance for comparisons: *p<0.05, **p<0.01, ***p<0.001, ****p<0.0001.

## Acknowledgements

This work was supported by the Northwestern University NUSeq Core Facility. Imaging work was performed at the Northwestern University Center for Advanced Microscopy generously supported by NCI CCSG P30 CA060553 awarded to the Robert H Lurie Comprehensive Cancer Center. Behavioral assays were performed at the Northwestern University Behavioral Phenotyping Core Facility. We thank Dr. Shoai Hattori for writing custom Matlab routines to analyze biophysics data and Dr. Corrina Burger for the generous gift of the CREB and control constructs. This work was supported by NIH grants R37 AG008796 (JFD), RF1 AG017139 (JFD), R01 NS063245 (JCPY), T32 AG020506 (DMC and XY), P30 AG13854 (DMC) and the Glenn/AFAR Scholarship for Research in the Biology of Aging (XY).

## Additional information

### Funding

| Funder | Grant reference number | Author |
|---|---|---|
| American Federation for Aging Research | Glenn/AFAR Scholarship for Research in the Biology of Aging | Xiao-Wen Yu |
| National Institutes of Health | T32 AG020506 | Xiao-Wen Yu<br>Daniel M Curlik II |
| National Institutes of Health | P30 AG13854 | Daniel M Curlik II |
| National Institutes of Health | R01 NS063245 | Jerry CP Yin |
| National Institutes of Health | R37 AG008796 | John F Disterhoft |
| National Institutes of Health | RF1 AG017139 | John F Disterhoft |

The funders had no role in study design, data collection and interpretation, or the decision to submit the work for publication.

### Author contributions

X-WY, Conception and design, Acquisition of data, Analysis and interpretation of data, Drafting or revising the article; DMC, MMO, JFD, Conception and design, Analysis and interpretation of data, Drafting or revising the article; JCPY, Conception and design, Analysis and interpretation of data, Drafting or revising the article, Contributed unpublished essential data or reagents

### Author ORCIDs

Xiao-Wen Yu, http://orcid.org/0000-0002-8974-0085
M Matthew Oh, http://orcid.org/0000-0002-6702-5785
John F Disterhoft, http://orcid.org/0000-0002-8817-7913

### Ethics

Animal experimentation: All procedures were performed in strict accordance with the recommendations in the Guide for the Care and Use of Laboratory Animals of the National Institutes of Health. All animals were handled according to protocols approved by Northwestern IACUC (Protocol

number: IS00002081, Animals Welfare Assurance: A3283-01) following NIH guidelines. All surgery was performed under isoflurane anesthesia, and every effort was made to minimize suffering.

## Additional files

**Supplementary files**

• Supplementary file 1. Young peak and slow post-burst AHP amplitudes do not differ. (a) Peak post-burst AHP amplitudes (mV) Aged CREB+ cells have smaller peak AHP's than CREB- and GFP, but young cell types do not differ from each other. Data represent mean ± SEM. *p<0.05, **p<0.01 compared to aged CREB+. (b) Slow post-burst AHP amplitudes (mV) Aged CREB+ cells have smaller slow AHP's than CREB- and GFP, but young cell types do not differ from each other. Data represent mean ± SEM. *p<0.05, **p<0.01 compared to aged CREB+.

• Supplementary file 2. Resting membrane potential and input resistance do not vary across cell types. (a) Resting Membrane Potential (mV) Resting membrane potential does not vary across different groups of cells. A two-way ANOVA revealed no significant effect of age ($F_{1,\ 63}$ = 0.6078, n.s.) or of cell type ($F_{2,\ 63}$ = 0.6122, n.s.). Data represent mean ± SEM. (b) Input Resistance (MΩ) Input resistance does not vary across different groups of cells. A two-way ANOVA revealed no significant effect of age ($F_{1,\ 63}$ = 0.8259, n.s.) or of cell type ($F_{2,\ 63}$ = 1.608, n.s.). Data represent mean ± SEM.

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
