## [Decision Letter]

Thank you for submitting your article "CREB overexpression in dorsal CA1 ameliorates long-term memory deficits in aged rats" for consideration by *eLife*. Your article has been reviewed by three peer reviewers, and the evaluation has been overseen by a Reviewing Editor and Gary Westbrook as the Senior Editor. The following individuals involved in review of your submission have agreed to reveal their identity: Tom Foster (Reviewer #2); Aline Marighetto (Reviewer #3).

The reviewers have discussed the reviews with one another and the Reviewing Editor has drafted this decision to help you prepare a revised submission.

Summary:

The reviewers were generally enthusiastic about the paper, although one reviewer had major reservations and requested an additional analysis, and all the reviewers requested substantial revisions for clarification.

Essential revisions:

1) Is there sufficient statistical power to detect the effects of CREB in young animals?

2) In the Discussion, the authors propose that one reason why they might not have seen memory enhancements in young CREB animals is due to the difficulty of the task for young animals. It would be helpful to determine whether CREB impacts memory in young animals in a more difficult version of the task.

3) The authors did not cite the first paper to have demonstrated the involvement of CREB in memory in mammals (Bourtchuladze et al., Cell 1994).

4) A concern is the implication that the effects were due to CREB activity and the idea that little is known about age-related changes in CREB activity. Throughout the manuscript, the language used to imply that the results were due to CREB activity should be modified to indicate that the results were due to increased expression. For example: However, it has yet to be tested if "increasing CREB activity also ameliorates age-related behavioral and biophysical deficits". This sentence should be altered. A review of the literature suggests that there is considerable support for the idea that induction of CREB activity (the level of pCREB) regulates memory in aged animals. Other examples include: Genoux et al., 2002; Porte et al., 2007; Countryman and Gold, 2007). The authors do not have to include these other references, but the point is that there is considerable evidence that learning induced pCREB or increasing pCREB during learning in aged animals improves memory and it has yet to be tested if "increasing CREB expression in aged animals will also ameliorates age-related behavioral and biophysical deficits".

5) In the Discussion, the authors may want to suggest that increasing CREB expression alters CREB activity. In this case, a statement suggesting the mechanism by which increased expression of CREB increases pCREB would be appreciated.

6) Speculation on how CREB expression influenced the AHP would also be appreciated. In discussing the idea that increased CREB expression influenced the AHP, the authors should consider the possibility that the change in the AHP is an epiphenomena due to CREB effects on behavior, rather than a direct effect of CREB expression. This point could be related to the work of the authors and others, which indicate that a decrease in the AHP is observed for young and aged animals that learn/remember on hippocampal-dependent tasks. The authors' data indicate a lack of effect of CREB on the AHP of young animals. Is it possible that the lack of effect on the AHP of young animals is due to the possibility that young animals were able to learn/remember on the hippocampal-dependent task regardless of CREB expression?

7) Similarly, the ability to modify the AHP in aged but not young animals (Discussion, fourth paragraph), a similar age specific effect is also noted for environmental enrichment which reduces the AHP in aged, but not young animals (Kumar and Foster, 2007) possibly due to differences in the ability to adapt to stress associated with behavioral experience.

8) Another major concern was that no evidence is provided that CA1 CREB expression (or activity) would diminish in aging. On the contrary, according to molecular analyses performed in the present study, there is, under control conditions, an age-related reduction in CREB mRNA levels in the DG (Figure 3) but not in the CA1 area (Figure 3). The reviewer was surprised to find no mention of these negative findings neither in the Results section nor in the Discussion. The lack of evidence for any age-related change in local CREB expression (or activity) has important implications for interpreting the positive effects of overexpressing CREB in the aged CA1. If CREB expression is not altered by aging, present rescue of age-related defects in memory and cellular excitability cannot reflect normalization of CREB activity/function, but might rather be due to the fact that overexpressing CREB can compensate for age-related changes in alternative cellular pathways. In that sense, the present study fails to elucidate critical molecular mechanisms involved in the age-related cognitive decline, and does not identify CREB as a key player. As one possible attempt to improve the paper and provide more convincing support to the authors' conclusion, it was suggested that the authors reanalyze molecular data presented in Figure 3) by dividing the Aged control group (N=17) into two subgroups: Aged Impaired and Aged Unimpaired (according to the performance in the Probe Trial performed 4 days after the last session of acquisition in the Morris water maze, i.e. just as done previously for behavioral analyzes in Figure 2). If a difference in CREB mRNA levels was to be found between the two aged groups (AI < AU), I would not disagree any longer with the conclusion on CREB being a key player in cognitive aging.

---

## [Author Response]

*Essential revisions:*

*1) Is there sufficient statistical power to detect the effects of CREB in young animals?*

Published behavioral studies with rats and mice, including those from our laboratory, have routinely reported findings with N’s of 5-10 animals per behavioral group. Examples include McKay et al. 2012, Matthews et al. 2009, Sekeres et al. 2010, and Josselyn et al. 2001. Therefore, we do believe that with N’s of 7-8 for our young groups, there is indeed sufficient statistical power to detect an effect of CREB, if it were present. We have also re-colored the young data points to differentiate between GFP and CREB data points in Figure 2, which shows near identical distribution, with substantial overlap of the two young groups.

*2) In the Discussion, the authors propose that one reason why they might not have seen memory enhancements in young CREB animals is due to the difficulty of the task for young animals. It would be helpful to determine whether CREB impacts memory in young animals in a more difficult version of the task.*

To fully answer this question, we trained a new group of AAV-GFP and AAV-CREB injected animals on a more difficult version of the water maze task (new Figure 4). The new data showed that in fact young animals were still not impacted by AAV-CREB, even on the more difficult version of the task, which removes the possibility of a ceiling effect causing the lack of facilitation. These results do not concur with studies showing facilitated behavioral performance in young animals using HSV virus, or mutant forms of CREB. We believe that the differing results can predominantly be attributed to the nature of the gene expression achieved by HSV vs. AAV. HSV tends to give expression that is short lasting and strong, while AAV is more subtle and stable. Given young animals are already in an “optimal state” we believe that stronger CREB overexpression, such as that achieved by HSV, or a mutant form of CREB is required to facilitate young behavior. This is discussed in the revised Discussion.

*3) The authors did not cite the first paper to have demonstrated the involvement of CREB in memory in mammals (Bourtchuladze et al., Cell 1994).*

We apologize for the oversight, which has been corrected. The citation has been added to the Introduction of the manuscript (third paragraph).

*4) A concern is the implication that the effects were due to CREB activity and the idea that little is known about age-related changes in CREB activity. Throughout the manuscript, the language used to imply that the results were due to CREB activity should be modified to indicate that the results were due to increased expression. For example: However, it has yet to be tested if "increasing CREB activity also ameliorates age-related behavioral and biophysical deficits". This sentence should be altered. A review of the literature suggests that there is considerable support for the idea that induction of CREB activity (the level of pCREB) regulates memory in aged animals. Other examples include: Genoux et al., 2002; Porte et al., 2007; Countryman and Gold, 2007). The authors do not have to include these other references, but the point is that there is considerable evidence that learning induced pCREB or increasing pCREB during learning in aged animals improves memory and it has yet to be tested if "increasing CREB expression in aged animals will also ameliorates age-related behavioral and biophysical deficits".*

We thank the reviewers for raising this important point. We have modified the text to replace “CREB activity” with “CREB levels” or “CREB expression” where appropriate.

*5) In the Discussion, the authors may want to suggest that increasing CREB expression alters CREB activity. In this case, a statement suggesting the mechanism by which increased expression of CREB increases pCREB would be appreciated.*

Upon the reviewers’ suggestion, we have included a section in the Discussion (second paragraph). Taking into consideration that CREB activation is thought to be decreased with aging (via increases in calcineurin and decreases in PKA), we suggest that increasing absolute amounts of substrate, CREB, would lead to increased levels of absolute pCREB.However, there have been suggestions that CREB activity may not rely solely on pCREB levels, this has also been acknowledged in the same section.

*6) Speculation on how CREB expression influenced the AHP would also be appreciated. In discussing the idea that increased CREB expression influenced the AHP, the authors should consider the possibility that the change in the AHP is an epiphenomena due to CREB effects on behavior, rather than a direct effect of CREB expression. This point could be related to the work of the authors and others, which indicate that a decrease in the AHP is observed for young and aged animals that learn/remember on hippocampal-dependent tasks. The authors' data indicate a lack of effect of CREB on the AHP of young animals. Is it possible that the lack of effect on the AHP of young animals is due to the possibility that young animals were able to learn/remember on the hippocampal-dependent task regardless of CREB expression?*

We thank the reviewers for raising this interesting point and have included a section in the Discussion to address it (eighth paragraph). While it is not currently understood how CREB might directly affect the AHP, we have now added our thoughts on the possible mechanisms at play. Importantly, we do not believe our observed effects on the AHP are driven by the animals’ behavioral performance, due to the time point at which we measured the AHP; 2-3 weeks after the last behavioral test. Previous biophysical studies have shown that learning-induced changes to the AHP return to baseline two weeks after the end of behavioral testing (Moyer et al., 1996; Thompson et al., 1996; Tombaugh et al., 2005). Furthermore, previously published studies in young animals indicate that increasing CREB activation or CREB overexpression reduces AHP size in neurons (Lopez de Armentia et al., 2007), which in turn makes those neurons more likely to be a part of a new memory engram (Han et al., 2007, Zhou et al. 2009, Yiu et al. 2014). Therefore, we believe that the observed behavioral facilitation was a result of the AHP size, rather the than other way around.

*7) Similarly, the ability to modify the AHP in aged but not young animals (Discussion, fourth paragraph), a similar age specific effect is also noted for environmental enrichment which reduces the AHP in aged, but not young animals (Kumar and Foster, 2007) possibly due to differences in the ability to adapt to stress associated with behavioral experience.*

We thank the reviewers for this suggestion. The citation has been added (Discussion, fourth paragraph).

*8) Another major concern was that no evidence is provided that CA1 CREB expression (or activity) would diminish in aging. On the contrary, according to molecular analyses performed in the present study, there is, under control conditions, an age-related reduction in CREB mRNA levels in the DG (Figure 3) but not in the CA1 area (Figure 3). The reviewer was surprised to find no mention of these negative findings neither in the Results section nor in the Discussion. The lack of evidence for any age-related change in local CREB expression (or activity) has important implications for interpreting the positive effects of overexpressing CREB in the aged CA1. If CREB expression is not altered by aging, present rescue of age-related defects in memory and cellular excitability cannot reflect normalization of CREB activity/function, but might rather be due to the fact that overexpressing CREB can compensate for age-related changes in alternative cellular pathways. In that sense, the present study fails to elucidate critical molecular mechanisms involved in the age-related cognitive decline, and does not identify CREB as a key player. As one possible attempt to improve the paper and provide more convincing support to the authors' conclusion, it was suggested that the authors reanalyze molecular data presented in Figure 3) by dividing the Aged control group (N=17) into two subgroups: Aged Impaired and Aged Unimpaired (according to the performance in the Probe Trial performed 4 days after the last session of acquisition in the Morris water maze, i.e. just as done previously for behavioral analyzes in Figure 2). If a difference in CREB mRNA levels was to be found between the two aged groups (AI < AU), I would not disagree any longer with the conclusion on CREB being a key player in cognitive aging.*

The reviewers raise an important issue. We reanalyzed the molecular data of the aged control animals as suggested, but did not find a difference in CREB mRNA levels between AI and AU animals (new Figure 3—figure supplement 1). This suggests that CREB mRNA levels are not related to behavioral performance. However, numerous studies have shown CREB protein levels/activation to be impaired with aging (Foster et al., 2001, Karege et al., 2001), more specifically, AI rats have less CREB protein than AU (Brightwell et al., 2004). Together, our present CREB mRNA results and previous CREB protein reports suggest that while CREB function is impaired at the protein level, CREB mRNA levels are not a biomarker of behavioral performance. Therefore, we also carried out western blots on previously frozen tissue from the aged GFP animals trained on the less challenging water maze protocol. Unlike previous reports on CREB protein, which either measured levels from naïve animals, or from trained animals shortly after training, we measured basal levels of CREB protein (2-3 weeks after the behavioral testing was completed). At this time point, we show that an animals’ basal levels of CREB protein is highly predictive of their behavioral performance – in this case on probe 4 (new Figure 3—figure supplement 2). Given this evidence from our new results and previous studies from others, we believe it is very likely our behavioral results were due to a normalization of impaired CREB function. However, as the reviewers correctly pointed out, we have not ruled out the possibility that CREB overexpression led to biophysical and behavioral benefits via other means. These points of discussion have now been included in the revised manuscript (Discussion, last two paragraphs).